# 3D-Scene-GAN: Three-dimensional Scene Reconstruction with Generative Adversarial Networks

**Chong Yu**
Software and Services Group
Intel
Shanghai, 200241, China
chong.yu@intel.com

**Young Wang**
Software and Services Group
Intel
Shanghai, 200241, China
young.wang@intel.com

## ABSTRACT

Three-dimensional (3D) Reconstruction is a vital and challenging research topic in advanced computer graphics and computer vision due to the intrinsic complexity and computation cost. Existing methods often produce holes, distortions and obscure parts in the reconstructed 3D models which are not adequate for real usage. The focus of this paper is to achieve high quality 3D reconstruction performance of complicated scene by adopting Generative Adversarial Network (GAN). We propose a novel workflow, namely 3D-Scene-GAN, which can iteratively improve any raw 3D reconstructed models consisting of meshes and textures. 3D-Scene-GAN is a weakly semi-supervised model. It only takes real-time 2D observation images as the supervision, and doesn't rely on prior knowledge of shape models or any referenced observations. Finally, through the qualitative and quantitative experiments, 3D-Scene-GAN shows compelling advantages over the state-of-the-art methods: balanced rank estimation (BRE) scores are improved by 30%-100% on ICL-NUIM dataset, and 36%-190% on SUN3D dataset. And the mean distance error (MDR) also outperforms other state-of-the-art methods on benchmarks.

## 1 INTRODUCTION

Three-dimensional (3D) Reconstruction is the killer technique in applications like surveying and mapping (Siebert & Teizer, 2014), medical imaging (Marro et al., 2016), 3D printing (Murphy & Atala, 2014), virtual reality (Sra et al., 2016), robotics (Guizilini & Ramos, 2016), etc. Due to the intrinsic complexity and computational cost, 3D reconstruction also arouses great interests of researchers in computer graphics and computer vision areas.

In this paper, we propose a 3D scene reconstruction workflow named 3D-Scene-GAN that combines latest GAN principle as well as advantages in traditional 3D reconstruction methods like SFM and MVS. By the fine-tuning adversarial training process of 3D scene generative model and discriminative model, the proposed workflow can iteratively improve the reconstruction quality. The main contribution of our work can be summarized as following items.

- 3D-Scene-GAN is a weakly semi-supervised model. It only takes real-time 2D observation images as the supervision, and has no reliance of shape priors, CAD model libraries or any referenced observations.
- Unlike many state-of-the-art methods can only generate voxelized objects or some simple isolated objects such as chair, car, plane, etc., 3D-Scene-GAN can be applied to generate very complicated 3D reconstructed scene, and still obtains decent result.
- 3D-Scene-GAN is compatible to any existing 3D reconstruction methods that represent reconstructed results with mesh and texture. This workflow can be applied to further improve mesh structure and texture quality of existing 3D objects reconstructed by other methods.

The implementation website of our project is available at https://github.com/dxxz/3D-Scene-GAN.

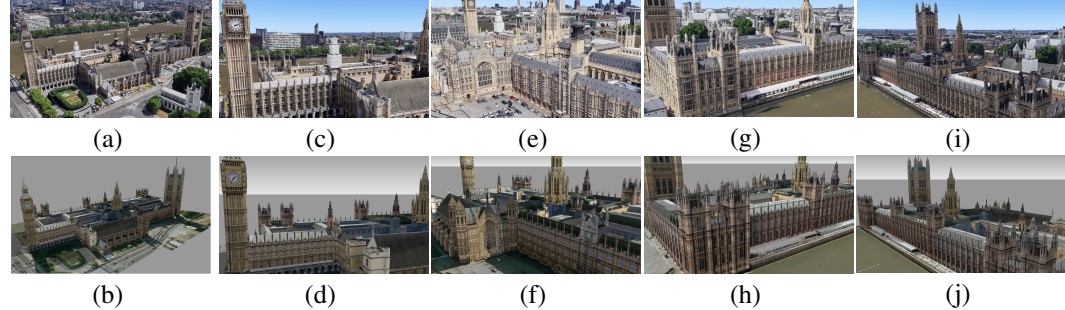

Figure 1: Illustration for 3D-Scene-GAN's principle. The Palace of Westminster is the target scene for 3D reconstruction. (a) real 3D scene. (b) reconstructed 3D scene. (c,e,g,i) are observed 2D images from real scene. (d,f,h,j) are observed 2D images from reconstructed scene. Each 2D scene image pair ((c)-(d); (e)-(f); (g)-(h); (i)-(j)) is observed with same position and viewpoint.

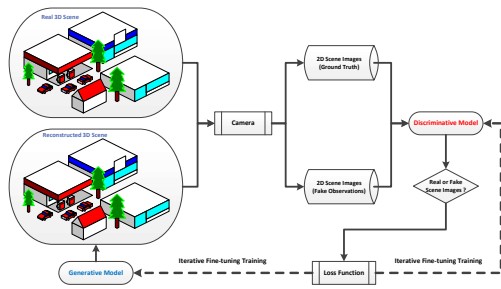

Figure 2: Framework and workflow chart of 3D-Scene-GAN.

## 2 PRINCIPLE OF 3D-SCENE-GAN

Imagine a person wants to discriminate the real scene and artificially reconstructed scene. So firstly, he observes in real 3D scene, then in reconstructed 3D scene. If this person observes the reconstructed 3D scene at exactly the same positions and viewpoints as in the real 3D scene, and all the observed 2D scene image pairs in reconstructed and real 3D scene are exactly the same. Then he can hardly differentiate reconstructed 3D scene from real 3D scene. Figure 1 illustrates this concept.

Combining the purpose of 3D scene reconstruction and GAN model, we propose the novel workflow, namely 3D-Scene-GAN. It consists of 3D scene generative model and discriminative model. Here, we can imagine the discriminative model as the observer aforementioned. The purpose of generative model is to reconstruct new 3D scene which is aligned with real 3D scene, and attempts to confuse the discriminative model. While the purpose of discriminative model is to classify reconstructed and real 3D scene. When 3D-Scene-GAN model achieves Nash Equilibrium, generative model can reconstruct 3D scene which exactly aligns with the character and distribution of real 3D scene. And at the same time, discriminative model returns the classification probability 0.5 for each observation pair of reconstructed and real 3D scene. This is also aligned with the evaluation criterion of 3D reconstructed scene. In conclusion, solving the 3D reconstruction problem is equal to making 3D-Scene-GAN model well-trained and converged.

## 3 3D-SCENE-GAN FRAMEWORK AND NETWORK STRUCTURE

The overall framework of 3D-Scene-GAN is shown in Figure 2. As 3D-Scene-GAN needs to get observed 2D scene images in reconstructed 3D scene, we import the reconstructed 3D model into Blender and OpenDR (Loper & Black, 2014). In Blender, a virtual camera is setup with same optical parameters as real camera, to collect 2D images along real camera trajectory. OpenDR is a differentiable renderer for mapping 3D models to 2D images, as well as back-propagating gradients of 2D images to 3D models. Differentiable renderer is necessary, because GAN structure needs to be fully differentiable to pass the discriminators gradients to update the generator.

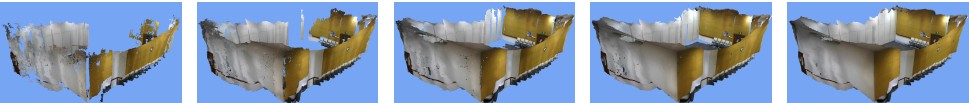

Figure 3: Network structure of 3D-Scene-GAN (above: generative, below: discriminative).

Figure 4: Reconstructed results of 3D-Scene-GAN. The reconstructed scene is an assembly hall in Intel site. Size of the hall is 23 meters in length, 11 m in width and 5 m in height.

For 3D-Scene-GAN, we apply 101-layer ResNet (He et al., 2016) as discriminative network. Because generative network needs to reconstruct the 3D model, so we change all convolutional layers to volumetric fully convolutional layers. Typical ResNet applies batch normalization to achieve stable training performance. However, batch normalization makes the discriminative model to map from a batch of inputs to a batch of outputs. 3D-Scene-GAN wants to keep the mapping relation from single input to single output. We replace batch normalization by layer normalization for generative and discriminative networks to avoid the correlations introduced between input samples. ReLU is replaced with parametric ReLU to improve the training performance. Moreover, to improve convergence, we use Adam solver instead of SGD solver. In practice, Adam solver can work with a higher learning rate when training 3D-Scene-GAN. The detailed network structures are shown in Figure 3.

## 4    EXPERIMENTAL RESULTS

Qualitative reconstructed results achieved in the iterative training process of 3D-Scene-GAN are shown in Figure 4. The learning rate of generative and discriminative networks is 0.063.

In quantitative experiment, we take Visual-SFM (Wu et al., 2011), SUN3D-SFM (Xiao et al., 2013), Kintinuous (Whelan et al., 2013), DVO-SLAM (Kerl et al., 2013), RRoIS (Choi et al., 2015) and ElasticFusion (Whelan et al., 2016) for comparation. Two metrics are used to measure reconstruction accuracy. One metric is the mean distance error (MDR) from each point on the ground-truth surfaces to the nearest point in reconstructed models. Balanced rank estimation (BRE) (Wauthier et al., 2013) is used as another metric to compute scores for all pairs of reconstructed scene models versus ground truth. BRE indicator is in the range of -1 to 1. If BRE is closer to 1, it means there is less relative difference between reconstructed scene and real 3D scene. The quantitative comparative results are shown in Table 1. According to MDE and BRE indicators obtained on ICL-NUIM and SUN3D datasets, 3D-Scene-GAN outperforms all state-of-the-art 3D reconstruction methods. The closest competitor is RRoIS. 3D-Scene-GAN has almost same MDE with RRoIS. However, it can still achieve the relative BRE score improvement over RRoIS as 30%-100% on ICL-NUIM, and 36%-190% on SUN3D dataset.

Table 1: MDR and BRE for evaluation multiple algorithms on ICL-NUIM and SUN3D datasets

| Dataset | | ICL-NUIMD | | | | SUN3D | | |
|---|---|---|---|---|---|---|---|---|
| Algorithms | Metrics | Living Room 1 | Living Room 4 | Office Room 3 | Office Room 4 | Dorm MIT-2 | Lab MIT | Studyroom MIT |
| Visual -SFM | MDR | 0.18 | 0.16 | 0.27 | 0.14 | 0.14 | 0.22 | 0.09 |
| | BRE | -0.13 | -0.19 | -0.32 | -0.26 | -0.4 | -0.1 | 0.26 |
| SUN3D -SFM | MDR | 0.09 | 0.08 | 0.13 | 0.08 | 0.11 | 0.11 | 0.07 |
| | BRE | 0.02 | -0.06 | -0.16 | -0.12 | -0.23 | 0.03 | 0.35 |
| Kintinuous | MDR | 0.22 | 0.14 | 0.14 | 0.13 | 0.11 | 0.13 | 0.11 |
| | BRE | -0.53 | -0.72 | -0.87 | -0.76 | -0.2 | -0.57 | -0.47 |
| DVO -SLAM | MDR | 0.21 | 0.08 | 0.11 | 0.1 | 0.07 | 0.11 | 0.1 |
| | BRE | -0.9 | -0.79 | -0.59 | -0.48 | -0.26 | -0.57 | -0.52 |
| RRoIS | MDR | 0.04 | **0.07** | **0.04** | 0.05 | **0.06** | 0.06 | **0.07** |
| | BRE | 0.47 | 0.55 | 0.6 | 0.23 | 0.1 | 0.22 | 0.5 |
| Elastic Fusion | MDR | 0.21 | 0.15 | 0.19 | 0.13 | 0.11 | 0.17 | 0.11 |
| | BRE | -0.33 | -0.54 | -0.63 | -0.57 | 0.01 | -0.34 | -0.28 |
| 3D-Scene -GAN | MDR | **0.03** | 0.08 | 0.05 | **0.04** | 0.08 | **0.05** | 0.09 |
| | BRE | **0.62** | **0.71** | **0.82** | **0.46** | **0.29** | **0.41** | **0.68** |

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
