# OpenReview forum: "3D-Scene-GAN: Three-dimensional Scene Reconstruction with Generative Adversarial Networks"
_ICLR.cc/2018/Workshop — Accept_

### Official Review · AnonReviewer2 · 2018-03-09
**Interesting idea but the description of the method lacks of clarity and details.**

**Rating:** 5
**Confidence:** 4

**Review:**

The idea of using GAN method to 3D scene reconstruction is novel and interesting. The proposed method performs well especially for the BRE Metric presented in Table 1.

Major concerns:
However, the method is presented as a 3D reconstruction technique, but it is not clear how the method can deal with a complete scene reconstruction from scratch. Indeed, an initial reconstruction (during the inference and the training) seems to be necessary and this aspect should be more discussed.
The method should be presented in more details:
•	How are the points of view chosen during the learning process? Is it possible to use multi-view images as input for the discriminator?
•	Are the real images segmented before being fed to the discriminator? Indeed, with the background it seems trivial to classify real/fake images.
•	The details concerning the generator are missing. In particular, how the rendering is achieved with respect to the generator error?
•	The method needs to compute the camera pose from the 2D image and the 3D model. However, the initialization robustness to the estimated pose is not discussed?

Minor concerns:
The writing of the paper can be improved:
•	To long sentences, for example, “If this person observes the reconstructed 3D scene at exactly the same positions and viewpoints as in the real 3D scene, and all the observed 2D scene image pairs in reconstructed and real 3D scene are exactly the same.”
•	Missing word, for example, “Unlike many state-of-the-art methods can only generate voxelized objects or some simple isolated objects such as chair, car, plane, etc., 3D-Scene-GAN can be applied to generate very complicated 3D reconstructed scene, and still obtains decent result.”
•	Figure 2 and 3: the text size is too small.
•	It could be more adequate to remove Figure 1 and to replace it with Figure 4 by adding 2D images of the real scene which are clearly missing and by presenting different viewpoints.

---

### Official Review · AnonReviewer4 · 2018-03-17
**Far from reproducible, would benefit from another edit**

**Rating:** 5
**Confidence:** 4

**Review:**

GANs are currently popular for synthesizing structured objects, the vast majority of attention has been on 2D images. This paper looks at a natural extension to 3D scenes, which is likely to be a promising area.

It isn't possible to reproduce the proposed method from the paper, and the linked github site doesn't fill in that detail. Figure 1 isn't actually created by the method? Even in this ideal case, the rendered 2D images would be easy to distinguish from the photos, so what constraints are preventing the classifier from rapidly "over-fitting"?

The quality of the paper could be improved. Figure 2 is too small to read. The text is quite hard to follow -- I think the description of the principle only makes sense because I already know what a GAN is. This text could be tightened up and more precise experimental detail given.

Minor: This part: "However, batch normalization makes the discriminative model to map from a batch of inputs to a batch of outputs. 3D-Scene-GAN wants to keep the mapping relation from single input to single output. We replace batch normalization by layer normalization for generative and discriminative networks to avoid the correlations introduced between input samples." -- is hard to follow, and the methods mentioned are not cited.

---

### Public Comment · ~Oriol_Vinyals1 · 2018-02-17
**Please Fix Length**

Your paper violates by a few lines the 3 page limit (see https://iclr.cc/Conferences/2018/CallForWorkshops). Please send us a fixed version of your PDF at iclr2018.programchairs@gmail.com by the end of Monday, February 19th, or else we will reject your paper.

Thanks,
ICLR2018 Program Chairs

---

> ### Public Comment · ~Chong_Yu2 · 2018-02-18
> **Submitted the fixed length version to ICLR2018 Program Chairs**
>
> Thanks for ICLR2018 Program Chairs. We submitted the fixed length version to iclr2018.programchairs@gmail.com on February 17th, 2018.

---

### Decision · Program_Chairs · 2018-03-20
**ICLR 2018 Workshop Acceptance Decision**

**Decision:**

Accept

**Comment:**

Congratulations, your paper was accepted to the ICLR workshop.